# Harnessing the Potential of Chitosan and Its Derivatives for Enhanced Functionalities in Food Applications

**DOI:** 10.3390/foods13030439

**Published:** 2024-01-29

**Authors:** Kexin Yi, Shiyuan Miao, Bixing Yang, Sijie Li, Yujie Lu

**Affiliations:** 1School of Grain Science and Technology, Jiangsu University of Science and Technology, Zhenjiang 212100, China; 220111801102@stu.just.edu.cn (K.Y.); shy.miao@just.edu.cn (S.M.); yangbixing2021@163.com (B.Y.); lisijie0807@163.com (S.L.); 2School of Biotechnology, Jiangsu University of Science and Technology, Zhenjiang 212100, China

**Keywords:** chitosan, extraction method, chemical modification, application in the food industry

## Abstract

As one of the most abundant natural polysaccharides that possess good biological activity, chitosan is extracted from chitin. Its application in the food field is being increasingly valued. However, chitosan extraction is difficult, and its poor solubility limits its application. At present, the extraction methods include the acid–base method, new chemical methods, and biological methods. The extraction rates of chitin/chitosan are 4–55%, 13–14%, and 15–28%, respectively. Different chemical modifications have different effects on chitosan, making it applicable in different fields. This article reviews and compares the extraction and chemical modification methods of chitosan, emphasizing the importance of green extraction methods. Finally, the application prospects of chitosan in the food industry are discussed. This will promote the understanding of the advantages and disadvantages of different extraction methods for chitosan as well as the relationship between modification and application, providing valuable insights for the future development of chitosan.

## 1. Introduction

Food safety has attracted the attention of researchers in recent years. The existing standards concerning food industry materials not only require good performance but also require less pollution [1]. At present, the raw materials used in the food industry are mostly non-degradable polymers, such as polypropylene and polythene. Therefore, the main trend present in food industry materials is the use of renewable natural resources, such as biopolymer chitosan (CS) [2]. CS is an abundant resource, and the subsequent preparation methods are simple, making it a favorable option [3].

CS is a linear amino polysaccharide consisting of D-glucosamine and N-acetyl-D-glucosamine units, which can be obtained via the deacetylation of chitin. CS can be extracted from the cell walls of fungi, the shells of crustaceans, and the stratum corneum of insects [4]. At present, the acid–base extraction method is mainly used to extract CS from crustaceans. The main steps include removing inorganic salts with a strong acid and removing proteins and acetyl groups with a strong alkali [5]. The extraction rate, deacetylation degree (DD), and molecular weight (MW) obtained from shrimp shells using traditional acid–base extraction methods were 4–55%, 51–98%, and 160–1050 kDa, and the extraction yield of chitin was 16–20%. The extraction rate, acetylation degree (DA), and MW of chitin from shrimp shells using new chemical extraction methods (ionic liquid, IL; deep eutectic solvent, DES) were 13–14%, 94–99%, and 56–312 kDa. The extraction rate and DA of chitin from shrimp shells using the biological extraction method were 15–28% and 83–96.33 kDa, respectively (Table 1). The extraction rate, DD, and MW are related to the source and extraction conditions. The MW and DD of CS affect its solubility and biological activity [6]. The traditional CS extraction process, which uses high concentrations of chemical reagents for a long time at high temperatures, can cause environmental pollution. The reagents used in the biological extraction method can be recycled, and they are environmentally friendly and can generate additional value-added products [7]. Therefore, pollution-free extraction methods are becoming increasingly popular. CS is soluble in acidic medium but not in water or organic solvents [4]. CS has good biocompatibility and degradability. In addition, CS also has a wide range of biological activities, such as antibacterial, antioxidant, immune regulation, antitumor, inhibiting fat absorption, lowering blood pressure, and cholesterol-lowering activities [8]. Therefore, its application in food, medicine, the chemical industry, textiles, and other fields has received great attention. However, CS has poor solubility in water and weak functional efficiency [8]. The hydroxyl and amino groups of CS make it easy to undergo chemical modification that can improve the solubility and biological activity of chitosan, thus expanding the application of CS and its derivatives in different fields [9].

The researchers of this article searched for keywords such as chitin, chitosan and its derivatives, and food preservation in the PubMed and Mendeley databases. The relevant papers of the past 10 years were summarized, and the relevant papers published in the past 5 years were described for the first time. This article compares and summarizes the extraction methods, chemical modifications, and applications of CS in the food industry, providing theoretical guidance for green extraction, chemical modification, and extended application of CS. 

## 2. Extraction Method of Chitosan

The traditional extraction process of CS is primarily divided into three stages: deproteinization, decalcification, and deacetylation. Some processes pursuing high-quality CS will add decolorization steps [32]. At present, the acid–base extraction method is the most widely used extraction technology in the industry owing to its low cost and simple operation. However, the large amount of waste liquid produced through the use of the acid–base extraction method causes serious environmental pollution [33]. In order to solve the problems of the acid–base extraction method, the pollution-free extraction process of CS has increased in popularity recently. Replacing acid–base reagents with green and safe chemical reagents or microorganisms is the focus of current research [34]. Biological extraction methods have attracted wide attention from scholars owing to their mild working conditions and low energy consumption. At present, a large number of studies have reported on the strain fermentation and enzymatic catalysis methods to extract CS [35]. At the same time, the emergence of new solvents such as IL and DES also show great application potential in the extraction of CS. The use of these new solvents is expected to become a key link in the green extraction process of CS [34]. The extraction methods of CS, including chemical methods and biological methods, are shown in Figure 1. The literature on the above extraction methods is briefly summarized in Table 1.

### 2.1. Chemical Extraction Methods

#### 2.1.1. Acid–Base Extraction Method

At present, the mature CS extraction process uses strong acids to remove calcium carbonate and strong bases to remove protein and acetyl groups. However, during the production process, this method produces a large amount of waste liquid, such as sulfuric acid and sodium hydroxide [36]. Some researchers have explored the feasibility of replacing strong acids and alkalis, hoping to obtain a mild chemical extraction process to reduce pollution and waste liquid. Some researchers used phosphoric acid and ultrasonic to extract chitin. After acid hydrolysis, water was added to precipitate high MV chitin, and then different concentrations of ethanol solutions were added to precipitate lower MV chitin [37]. Some researchers used glutamic acid to react with calcium carbonate to generate water-soluble calcium glutamate, which replaces strong acids for demineralization. Adding ammonium hydrogen phosphate to the waste liquid generates insoluble calcium hydrogen phosphate and glutamic acid, and glutamic acid can be recovered by adjusting the pH [38]. Although the improved acid–base extraction method can reduce the generation of waste liquid, it cannot fundamentally solve the problem of environmental pollution and product structural damage caused by this method. Therefore, scholars have turned their attention to finding green and safe chemical reagents to replace strong acids and bases and exploring green chemical extraction methods for CS.

#### 2.1.2. Ethylene Diamine Tetraacetic Acid-Assisted Extraction Method

The EDTA-assisted method removes inorganic salts by complexing saturated EDTA solution with calcium ions, but removing proteins requires a strong alkali [36]. Zhang et al. [39] used CO_2_, the cationic resin, HCl (CHI-HCl), and EDTA (CHI-EDTA) to remove inorganic salts from the deproteinized lobster shells. The results showed that the ash content of chitin was similar to the three methods, while the DA and crystallinity (CrI) of CHI-HCl were lower than those of CHI-CO_2_ and CHI-EDTA. Compared with the acid–base extraction method, the EDTA-assisted method has better working conditions and product quality. EDTA is easy to recycle, but its cost is high [40]. The EDTA-assisted method still uses chemical reagents for deproteinization and deacetylation, so it cannot avoid the drawbacks of the acid–base extraction method. Despite the EDTA-assisted method having reusable performance, it has disadvantages such as environmental damage and the high cost of waste liquid treatment. These disadvantages limit its large-scale application.

#### 2.1.3. Ionic Liquid Extraction Method

ILs are salts composed of organic cations and inorganic anions in liquid state below 100 °C [41]. Its performance depends on the combined use of ions and has a wide range of adjustability. In general, ILs composed of different cations and anions have different functions and properties, which can meet different needs [42]. The cations can dissolve crustaceans and insect shells, and the anions can destroy the hydrogen bonds between and inside chitin. Antisolvent is a reagent that makes chitin insoluble in ILs. Adding antisolvent can precipitate chitin [43]. The factors affecting the solubility of chitin include its CrI, MW, DD, and properties. The higher the MW and CrI, the poorer the solubility of chitin. The higher the DD, the better the solubility of chitin [44]. Tolesa et al. [22] investigated the extraction ability of ammonium ILs systems: diisopropyl ethyl ammonium acetate ([DIPEA][Ac]), diisopropyl ethyl ammonium propionate ([DIPEA][P]), and dimethylbutyl ammonium acetate ([DMBA][Ac]) from shrimp shells. The experimental results reveal that the ILs play a remarkable role in the extraction of chitin from shrimp shells with high selectivity. These ammonium-based ILs can be a promising green solvent to extract chitin from wasted shrimp shells and then convert it into CS. Setoguchi et al. [19] successfully extracted chitin from crab shells by deproteinizing with 1-allyl-3-methylimidazolium bromide (AMIMBr) and demineralization with citric acid. Wineinger et al. [45] compared the effects of two DES/IL pretreatment methods on the deacetylation efficiency of chitin from two insect sources (*Bombyx eri*, BE, and *Hermetia illucens*, HI) and shrimp shells. The results showed that the acetylation rate of CS obtained from BE larva by IL pretreatment (13–17%) was lower than DES pretreatment (18–27%). In addition, ILs can destroy the structure of chitin, leading to the deacetylation of chitin, so it can be used to extract the deacetylation process of CS extraction. ILs have good deproteinization, demineralization ability, and the potential for deacetylation. Compared with the traditional chemical extraction method, the IL extraction method does not involve chemical reagents, which greatly reduces the generation of waste liquid and environmental pollution. Compared with ordinary solvents, ILs have the characteristics of low volatility, high-temperature resistance, and recyclability, making them suitable as green solvents [46]. There is relatively little research on the deacetylation effect of ILs and the safety of extracts. Therefore, it is necessary to explore ILs with deacetylation activity and verify their safety, providing support for the industrial application of ILs extraction of CS.

#### 2.1.4. Deep Eutectic Solvents Extraction Method

DES is a deep eutectic mixture, which is mainly composed of hydrogen bond donors and hydrogen bond acceptors through hydrogen bond interaction. The charge delocalization caused by hydrogen bonds leads to the melting point of the mixture being much lower than the melting point of each component [47]. At present, DES can be divided into four types: organic or inorganic salt + metal chloride, organic or inorganic salt + metal chloride + hydrate chloride, metal chloride + Lewis acid, and anhydrous metal chloride + Lewis acid [48]. Sun et al. [49] proposed a novel natural deep eutectic solvents (NADESs) system consisting of lactic acid and choline chloride for extracting chitin. By optimizing the process conditions, chitin with a purity of up to 99.33% was prepared, which was comparable to the results of the traditional method. The free H in NADESs has been confirmed to be the main reason for decalcification. Hong et al. [21] prepared DES from choline chloride and four organic acids. The extraction of chitin from lobster shells using DES was evaluated. The research found that the purity of chitin extracted with DESs is related to the type of acid. The purity of chitin extracted with choline chloride–malonic acid was the highest. The MW of chitin is related to the type of acid and reaction temperature. Rodrigues et al. [50] evaluated the phytotoxicity of ChCl/organic acid-based DESs on wheat seed. They also explored the efficient recovery of chitin contained in brown crab shell residues by DESs. The results showed that ChCl/lactic acid-based DES has the least phytotoxic effects. It can obtain pure chitin (98%) with characteristics similar to commercial chitin in a shorter time. Therefore, the ChCl/organic acid-based DESs can serve as a low-phytotoxic alternative extraction medium for the extraction of chitin. DES has the advantages of reusability, biodegradability, non-toxicity, high thermal stability, low flammability, and easy preparation. It is a potential substitute for traditional organic solvents [51]. Compared with the traditional chemical extraction method, the DES extraction method produces a small amount of waste liquid, which is environmentally friendly and recyclable [48]. Compared with ILs, DES has higher degradability and safety [52]. In the chemical extraction method, the DES extraction method is the most promising green extraction method. At present, there are few studies using DESs to extract CS on a large scale, and there is a lack of research on its safety. Therefore, further research is needed to fill the gap between laboratory and industrial applications.

### 2.2. Biological Extraction Methods

#### 2.2.1. Strain Fermentation Process Extraction Methods

The fermentation extraction methods use the fermentation characteristics of specific microbial strains to achieve the effect of deproteinization or desalination [34]. Fermentation conditions of the strain are relatively simple, the overall process cost is lower than that of the chemical extraction method, and there is no large amount of waste to be treated. According to the actual extraction needs, different strains can be selected for combination to improve the yield. The fermentation extraction methods can improve the quality of chitin, increase the efficiency of deacetylation, and facilitate subsequent processing [7]. The fermentation extraction methods have the advantages of being environmentally friendly and having flexible operation. The methods have been widely considered and studied. At present, the fermentation extraction methods include the single-strain fermentation method and mixed-strain fermentation method.

The single-strain fermentation extraction method usually uses proteolytic enzymes by strains to remove proteins. There are two ways to remove inorganic salts: one is to use hydrochloric acid, and the other is to use the acidic environment or acid production required by the strains during the fermentation to reduce the overall pH value and achieve the effect of removing inorganic salts [53]. Ghorbel-Bellaaj et al. [53] used *Pseudomonas aeruginosa A2* to extract chitin from shrimp shells. Under the conditions of shrimp shell concentration of 50 g/L, glucose of 50 g/L, culture for 5 days, and inoculum optical density of 0.05, the demineralization rate was 96%, and the deproteinization rate was about 89%. This environmentally friendly method (biological treatment) can be considered an effective pretreatment to produce a high-quality chitin. Cahyaningtyas et al. [54] found that *Bacillus cereus HMRSC30* isolated from shrimp paste could secrete protease and chitosanase at the same time. Under the optimal conditions, the deprotection rate of chitin was 83% and the decalcifying rate was 40%. The fermentation of this strain can be used to produce chitin. The single-strain fermentation extraction method has the advantages that traditional chemical extraction methods cannot replace. It can adjust the fermentation strains used according to actual needs to achieve different extraction purposes. The working conditions are relatively warm, and the reagents used are cheap and environmentally friendly [55]. Compared with the traditional chemical extraction method, the extraction efficiency of the single-strain fermentation extraction method is relatively low, the quality of the product is difficult to control, and chemical reagents cannot be completely removed [34]. Some strains will also change and destroy the crystal form of chitin during the fermentation process, reduce the deacetylation efficiency of chitin, and increase the subsequent application cost. In addition, the single-strain fermentation extraction method has relatively high technical requirements due to the limitations of strain growth conditions, fermentation time, and other factors. It is difficult to form large-scale production at present, and its long extraction process also limits the large-scale practical application of the fermentation method [56]. With the continuous deepening of the research on single-strain fermentation technology, ultrasound, and microwave have been added to assist in this technology. This improves the yield while achieving the modification of the chitin structure, laying the foundation for subsequent processing [57,58]. In addition, the use of strains grown under acidic conditions and the integration of deproteinization and decalcification is also one of the future development directions of this technology.

There are two methods to prepare chitin using the mixed strain fermentation extraction method. One is the two-step fermentation method, which uses two strains with different growth conditions for the deproteinization and desalination process [34]. The growth conditions of the two strains are different, so it is necessary to sterilize and change the culture medium during the switching process between deproteinization and desalination [26]. The other is the continuous fermentation method, where the latter strain can sterilize the previous strain, avoiding repeated sterilization and replacement of the medium [36]. Zhang et al. [59] extracted chitin from co-bacteria of *Bacillus subtilis* and *Acetobacter pasteurianus*. The deproteinization rate was 94.5% and the decalcification rate was 92%. In addition, the low degree of polymerization of chitin is beneficial for subsequent deacetylation. Garcia et al. [60] used *Lactobacillus brevis* and *Rhizopus oligospora* to extract chitin from shrimp head and breast wastes with a deproteinization rate of 96.0% and a decalcification rate of 66.45%. They successfully obtained high-value-added products such as astaxanthin and protein from fermentation wastes. Liu et al. [26] used *Lactobacillus rhamnoides* and *Bacillus amyloliquefaciens*
*(BA01)* to extract chitin from shrimp shell powders (SSPs) by the two-step fermentation method process. The results showed the chitin obtained by fermentation maintained the excellent physicochemical and structural properties of commercial chitin. The mixed strain fermentation extraction method does not use chemical reagents. It avoids the generation of pollution waste liquid and the destruction of the chitin structure, making the overall structure of the product more stable. Moreover, the fermented waste has a high utilization value, which improves the utilization rate and increases income. Compared with the chemical and single-strain fermentation extraction methods, the mixed-strain fermentation extraction method has natural advantages in reducing environmental pollution and improving the quality of chitin [36]. In recent years, the continuous fermentation method has become the mainstream technology for the green extraction of chitin by the biological method. The continuous fermentation method not only reduces the contamination of miscellaneous bacteria but also makes the operation of the mixed strain fermentation extraction method simpler [34]. However, there is not much research on the use of mixed bacterial fermentation for the deacetylation extraction of CS. The research on the extraction of chitin by the mixed-strain fermentation extraction method is still limited in the laboratory-shaking bottle environment, which is not suitable for practical large-scale production applications. The low-intensity ultrasound can promote the metabolic activity of bacteria, stimulate growth during the fermentation process, and thus shorten the fermentation time [57]. Therefore, the future research trend of the mixed strain fermentation method is to screen suitable strains for deacetylation and turn to the construction of large-scale fermentation-integrated devices. Introducing physical auxiliary means such as ultrasound to improve the stability and efficiency of the fermentation process and control the extraction quality of chitin.

#### 2.2.2. Bio-Enzyme Extraction Methods

With the modern world focusing on environmentally friendly products, more and more chemical processes are being replaced by enzymatic methods. Unlike the fermentation extraction methods, the biological enzyme extraction methods remove the protein by directly adding purified protease. At present, the commonly used proteases include alkaline protease, pepsin, and papain [61].

Alkaline proteases comprise more than 50% of the total world enzyme production [62]. Under strong alkali and high-temperature conditions, alkaline proteases can still maintain high activity and stability. It has a wide range of sources, simple working conditions, and natural advantages in the industrial treatment [63]. Cui et al. [30] used alkaline protease and the shell of *Oratosquilla* malate to extract chitin. The results showed that the deproteinization and decalcification rates of chitin obtained by this method were 92.78% and 94.11%. The extracted chitin has a good porous structure with better crystallinity and thermal stability than those extracted by traditional chemical methods. Mhamdi et al. [64] isolated serine alkaline protease from a *Micromonospora strain S103*. When this enzyme was used in chitin extraction, the deproteinization rate was up to 93%. Hammami et al. [65] isolated a strain of protease-producing *Bacillus* from seawater, and the crude enzyme preparation made by this strain can maintain high activity under alkaline and high temperatures. When extracting chitin from shrimp waste, it was found that the enzyme can achieve the effect of deproteinization and decalcification in one step. The deproteinization rate was 76%, and the decalcification rate was 95.17%. The bioactive peptides in the enzymatic hydrolysate can be used as feed or fertilizer, increasing the economic value of the residual residue. Due to their abundant sources, mild working conditions, and stable properties, alkaline proteases have great application potential in the field of chitin extraction.

Pepsin is one of the main proteolytic enzymes in the digestive system. It is also the current mainstream commercial protease. Because its activity needs to be activated by contact with gastric acid, it is considered to be an acidic protease [66]. Pepsin is a protease used in many different applications. There are few studies on the direct application of pepsin to the CS extraction process. In many cases, it is used in an immobilized form to prevent the contamination of reaction products [67]. Pepsin is immobilized in the primary amine group, which has fewer primary amine groups and poor stability under alkaline conditions. This makes the immobilization of pepsin more complicated [68]. Li et al. [67] immobilized pepsin with tetramethoxysilane and 3-aminopropyl trimethoxysilane and then applied it to the purification of CS. The results showed that the deproteinization rates of pepsin and sodium hydroxide were 53.8–80.4% and 12.3–21.2%. After immobilizing protease with the mixed silane, the deproteinization efficiency was higher than traditional chemical methods, and only a slight degradation of CS occurred. It can recycle high-value-added products, including pigments and peptides. However, this method still requires the use of strong acids to remove Inorganic salts [68]. The acidic working environment of pepsin provides the possibility for the simultaneous deproteinization and inorganic salts. The application of immobilized enzyme technology reduces the problem of protein contamination, enhances the recyclability of protease, and reduces the production cost.

Papain is a non-specific cysteine protease extracted from papaya. It maintains high activity in a wide pH range and can be used to extract natural substances, such as CS [69]. Compared with the above two enzymes, papain is relatively simple and cheap to obtain, and the hydrolysis efficiency is stronger than ordinary non-specific enzymes. Therefore, it has great application potential in extracting chitin [34]. Gartner et al. [70] used papain to extract chitin from shrimp shells, and the results showed that part of the extracted chitin structure was damaged, and papain could hydrolyze proteins and deacetylate chitin at the same time, which would lead to the formation of chitosan. Gopalakannan et al. [29] explored the optimal conditions for chitin extraction by papain. The results showed that the deproteinization rate of chitin was 73.1% after for 72 h of enzyme treatment. The degree of acetylation was higher in the chitin prepared by the enzymatic method (19.4%) than that by the chemical method (17.2%), so it has great potential in the related application field of CS. Ding et al. [38] successfully extracted chitin from crab shells by glutamic acid–enzymatic hydrolysis. Glutamic acid can be recycled as a decalcification agent. The calcium carbonate in the crab shell is converted into calcium hydrogen phosphate by calcium glutamate, and the protein is converted into amino acids and polypeptides, which can be used as feed additives. ‘Glutamic acid–enzymatic hydrolysis’ is a relatively closed process with the advantages of mild reaction, greatly reducing the discharge of wastewater, waste gas, and waste residue, and a high comprehensive utilization rate of raw materials. Rohyami et al. [71] found that papain had the function of hydrolase in the process of chitin deproteinization. The concentration of papain affects the DD of chitin. Studies have found that papain can hydrolyze protein and deacetylate at the same time to obtain CS with high antioxidant and antibacterial activity [70,71]. Therefore, it is expected to be applied to the extraction of CS in the exploration of the development potential of CS in food preservation.

At present, the cost of pure enzymes is high, and their application in industrial production cannot meet the actual needs. The yield of chitin extracted by biological enzymes is generally low, and the quality is difficult to guarantee. Shrimp and crab contain various proteases, which can be used as a source of developing biological proteases and recycling waste. The enzyme immobilization can be reused, which can effectively reduce the cost. In the extraction process, ultrasonic, microwave, and other auxiliary means were added to improve the extraction rate and enhance the stability of quality.

## 3. Preparation and Application of CS Derivatives

After the incomplete deacetylation of CS, a rigid crystal structure is formed between the hydrogen bonds of amino and hydroxyl groups, which affects its water solubility. It can only be dissolved in a dilute acid solution with pH < 5.5, limiting its application [72]. However, the presence of chemically active hydroxyl and amino groups on the CS molecular chain makes CS easy to modify. The preparation methods of CS derivatives mainly include chemical modification, chemical modification, and enzymatic degradation [73]. The common chemical methods of CS modification include alkylation, quaternization, carboxymethylation, acylation, Schiff base, and graft copolymerization [74]. By introducing functional groups into the CS skeleton, some new properties of CS can be improved or given, thereby expanding the application range of CS [75]. Compared with CS, the physical and chemical properties of CS derivatives were improved, and the unique properties of CS were retained. CS derivatives have been widely studied due to their good physical, chemical, and biological properties such as high viscosity, low toxicity, and antioxidant [72]. The chemical modification of CS can give chitosan new properties and increase the diversity of CS derivatives, which is the most effective way to expand the application range of CS.

### 3.1. Alkylation Reaction

Alkylation reaction is the introduction of alkyl groups on the amino groups of CS to generate alkylated derivatives of CS. When amino and aldehyde react together, the reaction product is reduced with sodium cyanoborohydride (NaBH_3_CN) or sodium borohydride (NaBH_4_) to obtain alkylated CS (AC) derivatives. The direct reaction between CS and halogenated alkyl groups under alkaline conditions can also produce AC derivatives (Figure 2A) [76,77]. Kurita et al. [78] carried out the N-alkylation of CS with NaHCO_3_ as an alkylating agent and accelerator in the aqueous phase and obtained N-carboxymethyl and N-phenyl CS with a high degree of substitution. Dung et al. [79] reported the use of alkali halides and sodium bicarbonate to obtain N-carboxymethyl CS by N-alkylation. The results showed that the degree of substitution of the CS-NH_2_ group was almost 100%. Sun et al. [80] found that AC has no cytotoxicity to L929, and the cell viability was higher than the national standard, indicating that AC had good cytocompatibility. Zhang et al. [81] reported that ACGS20 prepared with an N-AC sponge and graphene oxide (GO) can promote the release of intracellular Ca^2+^ and stimulate platelet activation. The ACGS20 is a good candidate composition for safe and effective hemostatic dressings. At present, there are few studies and applications of AC in the food industry. The research of AC mainly focuses on the preparation of medical materials for hemostasis and wound healing [81,82]. AC has good mechanical properties, biological activity, cell compatibility, and hydrophilicity, so it is expected to be applied in the food field.

### 3.2. Acylation Modification

The acylation reaction is one of the earliest reactions studied in the chemical modification of CS. The amino or hydroxyl groups on the CS molecule are acylated with organic acid derivatives (such as anhydrides and acid halides), and an aliphatic or aromatic acyl group is introduced into the molecular chain to obtain an acylated CS derivative (Figure 2B) [83]. The acylation reaction disrupts the hydrogen bonds between CS molecules, leading to changes in the crystal structure, thereby increasing solubility and expanding the application range of CS [84]. Recently, a large number of studies have been conducted on the drug release, biological activity, and safety of acylated CS. Qing et al. [84] studied the effect of succinyl CS content on cell toxicity, and no obvious cytotoxicity was observed in all succinyl CS samples. Nanda et al. [85] developed a formulation of acylated CS coated paclitaxel-loaded liposomes. The results showed that the results showed paclitaxel injection; this formulation had higher drug capture efficiency, slower drug release, and better pharmacokinetics, biodistribution, and tumor uptake properties. These results confirmed the potential application of acylated chitosan coating for targeted drug delivery. Huang et al. [86] prepared the acylated CS with caffeic acid. The results showed that the modified CS had stronger antioxidant and antibacterial activities against *Escherichia coli*, *Staphylococcus aureus*, and *Candida albicans*. Pork treated with modified CS had a longer shelf life than pigs treated with CTS. The acylated CS derivatives have enhanced biocompatibility, anticoagulation, and blood compatibility. In addition, acylated CS derivatives do not cause inflammatory reactions in humans, so acylated CS can be used as a carrier or sustained-release agent in the field of medicine. The antibacterial and antioxidant activities of acylated CS are improved, and it can be used as a food preservation material.

### 3.3. Carboxymethyl Modification

Carboxymethyl CS is one of the most widely used CS derivatives. The carboxymethylation reaction is the reaction between CS molecules and chloroacetic acid under strong alkali conditions to obtain carboxymethyl CS. Under similar conditions, the carboxymethylation reaction of CS can also occur on hydroxyl and amino groups to obtain carboxymethyl CS (Figure 2C) [87]. Introducing carboxyl groups in CS can improve its water solubility, biocompatibility, and film-forming properties. The utilization value of CS has been improved and the application scope of CS has been expanded [88]. Zhou et al. [89] prepared carboxymethyl CS–amylopectin-edible film rich in galangal essential oil and characterized the physical properties, structure, and preservation effect of mango. The experimental results showed that the blend film had good compatibility and thermal stability, and it had an effective preservative effect on mango fruit. Yue et al. [90] synthesized a carboxymethyl CS/polyethylene oxide nanofiber membrane and tested its apparent structure, antibacterial activity, hydrophilicity, and air permeability. The results showed that the CS nanofiber membrane had significant air permeability and antibacterial activity, which could effectively avoid water loss in strawberries and had a significant effect on extending the shelf life of strawberries. As the degree of carboxylation increases, the antibacterial performance of carboxymethyl CS first increases and then decreases. It exhibits good antibacterial performance within a wide range of carboxylation degrees, improving the utilization value of CS [91]. At present, research on carboxymethyl CS mainly focuses on the synthesis of nanofiber materials from plant essential oils and metal oxides as well as their application in the field of food preservation. Carboxymethyl CS composite materials have good compatibility, thermal stability, antibacterial activity, hydrophilicity, and permeability, and they have a preservation effect on fruits, vegetables, and meat [92].

### 3.4. Quaternization Reaction

The quaternary ammonium CS is divided into two categories. One is the synthesis of the quaternary ammonium salt of halogenated CS by the reaction of amino and alkyl in the CS [1]. Iodine-substituted alkanes are commonly used halogenation reagents due to their high reactivity. The other is the reaction of quaternary ammonium salt containing epoxy alkanes and CS to obtain quaternary ammonium salt of CS (Figure 2D) [93]. Nian et al. [94] prepared multifunctional packaging biopolymer-based films with modified metal–CS quaternary ammonium salt–gelatin and studied the characteristics of packaging and its anti-corrosion effect. The results show that the physical and chemical properties of the base membrane improved, and it has high ethylene adsorption performance. It has an obvious antibacterial effect on *Escherichia coli* and *Staphylococcus aureus* and can prolong the shelf life of perishable food. Pan et al. [95] synthesized (5-carboxy) (triphenyl) hydrogen bromide (HA-CS-NP) quaternary ammonium CS and prepared a multifunctional food packaging composite film. The film has good thermal stability and antibacterial function, has a good inhibitory effect on mango and papaya spores, and prolongs the fruit decay time. Presently, research on quaternary ammonium CS mainly focuses on preparing different multifunctional biopolymer membranes by combining them with other substances. Compared with CS, quaternary ammonium CS significantly improves its water solubility and antibacterial properties, and the composite material has good thermal stability and antibacterial function. Quaternary ammonium CS maintains the film-forming properties, biocompatibility, antibacterial and antioxidant properties of CS, and it also has excellent water solubility. It has extensive application value in fields such as food, medical materials, and textile processing.

### 3.5. Graft Copolymerization Reaction

CS introduces polymer side chains and their grafting copolymerization, such as alcohol, ester, acid, amide, etc. The grafting copolymerization reaction gives CS new excellent properties [96]. There are two main methods for CS grafting copolymerization: one is to generate macromolecular free radicals on the polymer skeleton, which triggers monomer polymerization. The second is the coupling between the active functional groups on the polymer molecular chains mentioned above and other polymer molecular chains (Figure 2E) [97]. Bi et al. [98] prepared a CS–graft–polyvinyl alcohol (PVA) membrane with an encapsulation efficiency of more than 95% for procyanidins and excellent long-term sustained release performance. Compared with traditional CS and CS–blend–PVA membranes, the CS–graft–PVA membrane has better mechanical properties and barrier properties and has ideal antibacterial activity and biofilm inhibition against foodborne pathogenic microorganisms and spoilage bacteria. Sabba et al. [99] prepared the graft copolymer of CS and acrylonitrile. The results showed that it had antibacterial activity against *Streptococcus pneumoniae*, *Staphylococcus aureus*, and *Escherichia coli*. Additionally, it also had antifungal activity against *Aspergillus fumigatus* and *Candida albicans*. Hassan et al. [100] prepared CS quaternary ammonium salt derivatives grafted with poly [2-(allyloxy) ethyl trimethyl ammonium chloride] or poly acryloyl trimethyl ammonium chloride (pATC). The results showed that pATC-grafted CS had higher antibacterial activity against *Aspergillus fumigatus* and *Aspergillus brasiliensis*. By graft copolymerization to modify CS, new functional polymer chains can be introduced. CS has been endowed with new excellent properties, and it maintains the original excellent performance [96]. Compared with CS, grafted CS has better antibacterial activity and physicochemical properties, making its application more extensive.

### 3.6. Schiff Alkalization Reaction

The carbonyl groups of aldehydes or ketones can effectively couple with the -NH_2_ group of CS, forming CS Schiff bases (CSSB) with imine characteristics (Figure 2F) [101]. CSSB changes the molecular structure of CS, enhances its hydrophilicity, and increases the number of positive ions. Compared with CS, CSSB has a better antibacterial activity [102]. Therefore, CSSB is considered one of the best choices to enhance the antibacterial activity of CS. Zhang et al. [103] prepared amphiphilic CS/iodine and polyaminoethyl CS citronellal Schiff base iodine (PACSC-I) composite films. The composite films have good hydrophilicity, water absorption, elasticity, biocompatibility, thermal stability, and antibacterial effect. Qin et al. [104] prepared CS/tannic acid (CTA) and CS/oxidized tannic acid (COTA) composite membranes by the tape method. The results showed that the composite film had excellent mechanical properties and antibacterial properties, and it had an obvious antibacterial effect on Escherichia coli. CTA film has broad application prospects in food packaging. Fontana et al. [105] investigated the antimicrobial activity of a novel CSSB. The results showed that CSSB significantly increased its antimicrobial activity against Gram-positive and Gram-negative bacteria and yeasts. According to literature reports, CSSB serves as a carrier for various substances [106], an adsorbent for various harmful substances [107], and a biomaterial with various biological activities [103]. The biological activity and adsorption capacity of CSSB are both attributed to CS [102]. These performance improvements provide prospects for the application of CSSB in fields such as biology, the food field, catalysis, sensors, and water treatment.

## 4. Applications of CS and Its Derivatives in the Food Field

CS is obtained from chitin and is considered one of the most abundant natural polysaccharides. In the last half-century, the application of CS in food preservation has received increasing attention due to its functional activity. Compared with earlier research, recent research has increasingly focused on the exploration of preservation mechanisms and more efficient targeted inhibition. This is due to the availability of more active composite ingredients and integration of more technologies, which has been gradually perceived as a “CS-based biofilm preservation” [2]. CS can increase the nutritional value of food, maintain food quality, and have a wide range of applications in the food field (Figure 3).

### 4.1. Application of CS and Its Derivatives in Food Preservation

Natural polymers have attracted much attention due to the huge environmental impact of the non-biodegradability of plastic-based packaging materials. The development of biopolymers from industrial waste can not only reduce waste but also lead to new food packaging materials. CS is a natural preservative, an alternative to synthetic polymers, and a raw material for new materials. It can be extracted from shrimp, crab shells, and silkworm pupal skin produced in seafood production. CS can be combined with plasticizers, essential oils, polysaccharides, proteins, and other components to prepare composite materials, which are suitable for food preservation and packaging. CS-based composites can preserve food by electrostatic spraying, coating, soaking, and other methods, and it can also extend their service life while maintaining the quality and characteristics of food and the biodegradability of polymers.

The CS composite materials can form a membrane with selective permeability to O_2_, CO_2_, and C_2_H_4_ in fresh fruits and vegetables. This film can effectively reduce the respiration intensity of fruits and inhibit ethylene production, browning, and enzyme activity, thereby maintaining the color, luster, and hardness of fruits and vegetables as well as extending their shelf life [108]. Jiang et al. [109] investigated the effect of electrostatic spraying (ES) technology on spraying CS strawberry quality during cold storage. The results show that the CS coating of ES forms a more continuous and uniform protective layer compared with conventional spraying. The CS coating effectively improves the overall quality of strawberries and prolongs shelf life. The CS coating of ES has great potential for industrial application of fruit preservation as a safe, inexpensive, and efficient technique. Cheng et al. [110] used CS–catechin coating to extend the storage time of Satsuma orange. The results showed that CS–catechin coating treatment significantly reduced the decay caused by *Penicillium citratum* and *Aspergillus Niger*. Bian et al. [111] grafted N-carboxymethyl CS with gallic acid and coated strawberries. The results showed that the weight loss rate of strawberries decreased from 12.7% to 8.4%, the titratable acidity content remained above 60%, and the decay rate decreased from 36.7% to 8.9%. The antioxidant capacity and nutritional components of strawberries were protected, and the shelf life was effectively extended. Olawuyi et al. [112]. coated fresh-cut cucumbers with different concentrations of edible CS solution and placed them in modified atmosphere packaging. Studies have shown that the quality retention rate of cucumber after CS coating is improved and carbon dioxide production is reduced. The concentration of CS solution had a significant effect on the performance of cucumber in modified atmosphere packaging. Qiu et al. [113] used high MV CS (H-CS), low MV CS (L-CS), and carboxymethyl CS (C-CS) coating to preserve green asparagus. The results found that L-CS and H-CS could effectively inhibit the growth of postharvest green asparagus *Fusarium*, and the inhibitory effect on spore germination and germ tube elongation was better than the C-CS. CS showed antifungal activity and an ability to stimulate some defensive responses during storage. CS also has an excellent antibacterial effect. Its attachment to the surface of the fruit can not only inhibit the growth of fungi on the surface of the fruit but also enhance the mechanical strength of the fruit, reduce damage, and provide a good antiseptic and fresh-keeping effect.

The storage methods for meat, aquatic products, eggs, and dairy products mainly include refrigeration, freezing, and cooked food processing. Meat, eggs, and dairy products have high levels of fat and fatty acids, which can easily oxidize and cause food spoilage, shortening the shelf life of meat products. Additionally, many harmful substances will be generated during the storage process [114]. To preserve these foods, antioxidants are usually required to form chelates with free iron ions in food, thereby inhibiting the catalytic activity of iron ions and exerting antioxidant effects [2]. A large number of studies have verified the good antioxidant activity of CS. Huang et al. [86] prepared acylated CS using caffeic acid and preserved pork. Compared with natural CS, modified CS showed stronger antioxidant and antibacterial activities against *Escherichia coli*, *Staphylococcus aureus*, and *Candida albicans*. Pork treated with modified CS had a longer shelf life. Song et al. [115] successfully prepared a cellulose–CS–citric acid membrane (C-Ch_x_-F). The C-Ch_x_-F film has excellent antibacterial properties and excellent decontamination performance in meat preservation. Chang et al. [116] found that a 95% deacetylated CS (DD95) shell reduced the heat resistance of *Clostridium perfringens* CCRC 10648 and CCRC 13019. In pork sausages, the addition of DD95 CS significantly slowed down bacterial proliferation and the production of volatile basic nitrogen. DD95 CS can reduce the concentration of sodium nitrite required for the color retention of pork sausage. Wu et al. [117] investigated the effect of CS, CS oligosaccharides, and glutathione on the freezing preservation of *Penaeus vannamei*. CS coating treatment can effectively inhibit the growth of bacteria, reduce total volatile basic nitrogen, and malondialdehyde, basically maintain the sensory characteristics of white shrimp during partial freezing storage, and prolong the shelf life of *Penaeus vannamei*. Karsli et al. [118] studied the effect of high MV (800 kDa) CS coating on the quality of catfish fillets after freezing at −20 °C for 6 months. The results showed that 3% coating could effectively inhibit microbial growth, control lipid oxidation, and reduce drip loss and cooking loss. Maintaining the color and texture of catfish fillets during frozen storage and non-irritating aspartic acid could replace irritating acetic acid. Yang et al. [119] prepared CS and syringic acid (SA) composite film and used it for the preservation of quail eggs. The results showed that CS-SA film had an obvious antibacterial effect on experimental bacteria and had a good preservation effect on quail eggs. Alfaifi et al. [120] successfully designed a water-soluble CS (WSC) and 2-acid propionic acid (APA) conjugate (WSC-APA) as a new safe multifunctional biological preservative. The minimum inhibitory concentration of WSC-APA could completely inhibit the proliferation of *Escherichia coli* O157:H7 and *Staphylococcus aureus* in refrigerated milk samples within 20 and 24 h. In addition, *Staphylococcus aureus* was not detected in refrigerated milk samples treated with WSC-APA (0.25 mg/mL) after 6 days of storage, and the number of coliforms decreased by 99.7% within 10 days. WSA-APA can be safely used without sacrificing its sensory and nutritional value to extend the shelf life of milk. The combination of CS and other natural antibacterial agents can ensure food safety and prevent the invasion of pathogenic microorganisms. The film prepared by CS has good mechanical properties and preservation effect on meat, seafood, eggs, and dairy products without affecting the sensory and flavor of the food. These factors have promoted the development of CS as a food preservation material.

The research of CS on food preservation is mostly carried out under refrigerated and frozen conditions. There is no research on the application of CS in the preservation of the same food under different conditions. The effect of the change in food quality on the properties of CS composites has not been studied. The addition of the above two areas of research aims to refine collaborative endeavors and technology aimed at refining CS applications across different environmental settings. Understanding how variations in raw material quality might influence CS performance and exploring strategies to ensure uniform preservation outcomes could significantly contribute to the broader discourse on food preservation methods.

### 4.2. Application of CS in Food Flocculan

CS is the only positively charged polymer in nature. The positively charged amino groups on CS molecules can interact with negatively charged anionic electrolytes in liquids. Its long-chain linear structure can adsorb multiple impurities on the same molecule, forming flocs, thereby disrupting the colloidal structure that plays a stabilizing role [121]. CS can be used as a flocculant for solid–liquid separation, removing suspended solid particles from liquids, or separating suspended solid products from solutions. CS can also be used as a clarifying agent to purify drinking water and clarify liquid beverages [122]. CS can increase the transparency of liquids and improve product yield and quality. It is an excellent flocculant and clarifying agent. It has been applied in the food industry. Yavaşer et al. [123] prepared CS cryopreserved material of polyhydroxyethyl methacrylate and immobilized it on papain. The results showed that the modified CS had a good clarification effect on apple juice, and it had no significant effect on the nutritional components of apple juice. The modified CS has reusability and storage stability. Compared with the traditional clarification process and CS, modified CS as flocculation material has a better clarification effect, stability, and efficiency, and it has natural advantages such as being non-toxic and degradable [124]. Venkatachalapathy et al. [125] found that CS can remove residual pesticides in the process of grape juice clarification. It was observed that the CS (0.05%) showed a higher pesticide removal efficiency (72%) when compared with other clarifiers. Compared to gelatin (57%), casein (47%), and bentonite (33%), activated carbon showed better pesticide removal efficiency (68%) at 0.05% for 4 h clarification. Tastan et al. [126] evaluated the application of CS as a clarifying agent in pomegranate juice production for the first time and investigated its effect on the quality characteristics of pomegranate juice. Compared to using bentonite and gelatin, pomegranate juice clarified with CS has comparable quality characteristics; therefore, using CS clarification is an equivalent method to traditional clarification. CS used as a clarifying agent does not affect the flavor and nutritional characteristics of the juice, and it maintains biological stability during the clarification process. Further study on the quality characteristics and storage potential of CS can promote its wide application in the juice industry.

### 4.3. Application of CS in Functional Foods

CS and its derivatives are used as functional food additives due to their diverse biological activities and harmlessness to the human body. The NH^2+^ of CS can capture negatively charged fats and Cl- in the human body, and it can also bind bile acids by electrostatic interaction. Low MV CS can even penetrate and repair islet cells, playing an important role in lowering blood lipids, blood glucose, and cholesterol, and stabilizing blood pressure [127]. Liu et al. [128] compared the effects of carbon monoxide (CO) and high MV CS (HC) on hepatic lipogenesis and lipid peroxidation, lipolysis, and intestinal lipid absorption in high-fat (HF) diet rats. Research has found that both HC and CO could reduce the biosynthesis of liver lipids in HF diet rats, but the effect of HC had a better effect on improving lipid accumulation in the liver of HF diet rats than CO. CS has lipid-lowering and weight loss effects and can reduce the absorption of lipids in the intestine. Liu et al. [129] studied the effects of COS on glucose metabolism and hepatic steatosis. The results showed that plasma glucose, urea, creatinine, uric acid, triglyceride (TG), total cholesterol (TC), and liver glucose-6-phosphatase activity were significantly decreased in diabetic rats. COS can alleviate abnormal glucose metabolism in diabetic rats by inhibiting gluconeogenesis and lipid peroxidation in the liver and SGLT2 expression in the kidney.

CS and its derivatives can selectively penetrate the membrane of cancer cells, interfere with the synthesis of DNA, RNA, protein, or enzyme, and disrupt hormone biosynthesis to inhibit the growth of cancer cells. It also shows anticancer activity through cellular enzymatic processes, anti-angiogenesis, immune enhancement, antioxidant defense mechanisms, and apoptotic pathways [75]. The lymphocytes in the human body are very sensitive to changes in the pH of the internal environment. CS and its derivatives can make the pH of body fluid tend to be weak and alkaline, thereby improving the activity of lymphocytes, enhancing the activity of lymphocytes to destroy cancer cells, and playing a role in cancer prevention [130]. Anushree et al. [131] evaluated that phosphorylated galactose-CS (PGC) had good metal chelation, iron ion reduction, and free radical scavenging activity. The number of tumor-free animals in the PGC treatment group increased, and the tumor diversity decreased significantly. This indicates that PGC has the potential for anti-cancer therapeutic. Dziadek et al. [132] used 2,3,4-trihydroxybenzaldehyde (THBA) to crosslink CS-based hydrogels and modified them with pectin (PC), bioactive glass (BG) and romamelanic acid (RA). Studies have shown that the substance has high antioxidant activity and an anti-proliferative effect on cancer cells but has no cytotoxicity to normal cells. Studies have demonstrated that CS and its derivatives have good metal chelation, iron ion reduction, and free radical scavenging activities. They have high antioxidant activity and anti-proliferation effects on cancer cells but have no cytotoxicity on normal cells, which shows the potential of anticancer therapy [131,132]. Some studies have shown that CS and its derivatives regulate lipid metabolism and glucose metabolism in the body, reduce intestinal absorption of lipids, and have lipid-lowering and anti-obesity effects [128]. CS oligosaccharides can inhibit gluconeogenesis and lipid peroxidation in the liver and the expression of SGLT2 expression in the kidney, thereby alleviating abnormal glucose metabolism in diabetes patients [129]. CS and its derivatives have been shown to have hypoglycemic, lipid-lowering, and anticancer effects, and they are non-toxic to human cells. Therefore, it can be used as an additive to regulate body immunity, improve the intestinal environment, and lose weight in food.

### 4.4. Application of CS in Food Industry Wastewater

CS can form a crosslinked polymer with heavy metal ions and benzene hazardous wastes in water, which has a good removal effect. CS is non-toxic, odorless, biodegradable, and does not cause secondary pollution [133]. It is also used to treat wastewater from the food industry and recover proteins. The recovered protein can be used as animal feed to increase profits and reduce pollution [134]. Sun et al. [135] prepared two multifunctional nano-CS flocculants by grafting a cationic monomer with carboxymethyl CS (CMCTS) to remove composite pollutants. The results showed that the nano-flocculants prepared by modified CMCTS with different structures had an ideal flocculation removal effect on dye wastewater and heavy metal wastewater. Han et al. [136] combined polyethyleneimine with CS to prepare a CS-based microsphere biosorbent (CP). The time required for CP to reach adsorption equilibrium was significantly shorter than for CS microspheres, and the adsorption rate was significantly improved. In addition, CP has obvious selectivity, and it can effectively remove Cr from printing and dyeing wastewater. Croitoru et al. [137] developed a composite membrane with CS and GO as adsorbents to remove inorganic pollutants in aqueous solution, such as heavy metal ions, especially Pb^2+^. The results showed that the CS/EDTA/GO membrane had broad application prospects as an effective adsorption material for removing heavy metal ions in water. Velasco et al. [138] prepared a biodegradable CS-based bio-composite material. The results showed that the composite could remove copper ions from aqueous solutions, and the adsorption capacity of the material was lower than its saturation. Ibrahim et al. [134] synthesized CS-cellulose (CS-CEL) and CS powdered activated carbon (CS-PAC) nanocomposites. They serve as green biodegradable clarifying agents for the treatment of carbonized bagasse in sugar production. Compared with the traditional clarification process, CS-PAC and CS-CEL as clarifiers improved the decolorization rate and reduced the turbidity. As an environmentally friendly biosorbent and flocculation material, CS-CEL and CS-PAC nanocomposites have high efficiency in sugar production. CS and its derivatives can be synthesized into nanocomposites with activated carbon, cellulose, and other substances. As an environmentally friendly biosorbent and flocculation material, the adsorption and decolorization efficiency are significantly improved. The developed composite materials have obvious adsorption selectivity for harmful substances, but the adsorption range is not wide enough. Some materials cannot be reused. It is important to expand the application range of flocculants in the food industry and improve the universality of the adsorption range and recyclability. Therefore, the covalent immobilization of CS derivatives and adsorbed substances will be a future development direction, which can not only enhance the flocculation clarification effect but also recycle, reduce costs, and improve efficiency.

## 5. Conclusions and Prospects

At present, CS is still prepared by chemical extraction methods in the industry. The use of acid and alkali will produce large amounts of industrial wastewater, increase costs, and pollute the environment. Numerous studies showed that the physicochemical characteristics of CS prepared by biological extraction methods were better than those of chemical extraction methods. The biological extraction methods are beneficial to the environment. Recycling by-products can increase revenue. Therefore, biological extraction methods are considered a promising technology. In addition, economical and green preparation methods should be explored to promote the large-scale production of CS. The commercial scale of CS sustainable production using green extraction technology will be a future research direction.

In recent years, the preparation of CS derivatives has made great progress. The chemical modification of CS is achieved by changing the reaction conditions to introduce the target group into the amino or hydroxyl group to obtain CS derivatives. The water solubility and biological activity of CS have been improved by chemical modification methods, especially antibacterial, antioxidant, antitumor, and hemostatic activities. Although the chemical modification methods of CS have radically matured, the reaction reagents and conditions in the chemical modification process are relatively harsh and harmful to the environment, and they may also affect human health. Therefore, it is necessary to find green and safe modification methods and develop stable CS resources.

CS and its derivatives have good physicochemical properties and biological activities. In addition to its application in the food industry, it can also be used in fields such as the environment, agriculture, pharmaceuticals, medical products, textiles, and cosmetics. In the food industry, CS is mostly used for food preservation. CS has good film-forming and degradability, but its antibacterial and antioxidant activities are relatively low compared to chemical preservatives. Biopolymers such as proteins, polyphenols, and peptides do not possess film-forming properties. Therefore, CS often binds to naturally occurring polymers with biological activity. Composite materials in different states are prepared according to actual needs, such as composite membranes, hydrogels, nano-lotion, and nanoparticles. These have expanded the application scope of CS and its derivatives. Compared with petroleum-based polymers with similar properties, CS and its derivatives have higher development costs, and some active mechanisms are not fully understood. This is the main reason why the development of new applications of CS and its derivatives is relatively slow and has not yet achieved large-scale industrial applications. Therefore, in addition to strengthening the development of CS modification technology, it is also necessary to strengthen the application research of CS derivatives. In summary, although the application of CS in certain fields is controversial, it has been proven to be a multifunctional natural polymer that has been applied in many fields.

## Figures and Tables

**Figure 1 foods-13-00439-f001:**
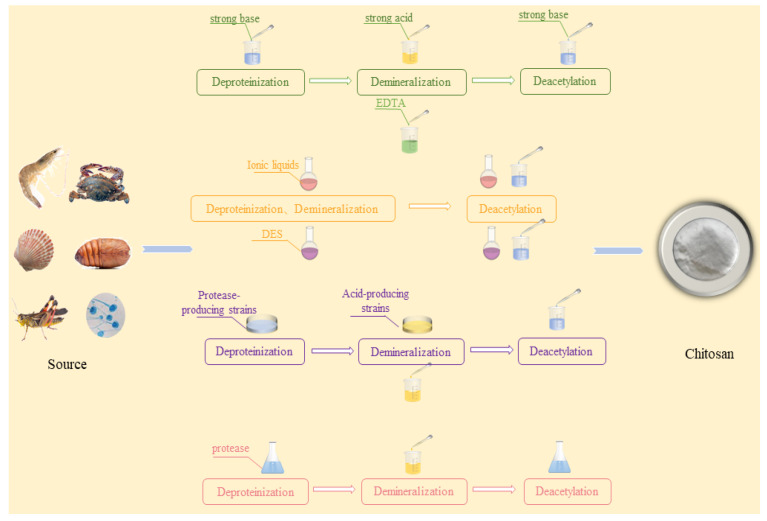
Process diagram of chitosan extraction methods.

**Figure 2 foods-13-00439-f002:**
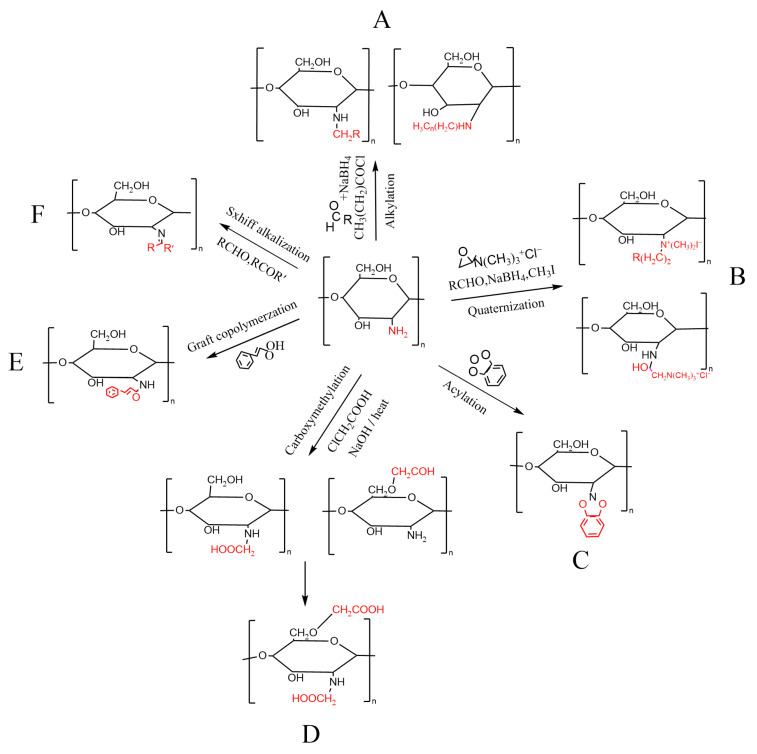
Structure and chemical modification of CS. (**A**) CS reacted with haloalkanes, aldehydes, NaBH_3_CN or NaBH_4_ to give AC; (**B**) CS was reacted with haloalkanes and quaternary ammonium salts containing epoxy alkane to obtain CS quaternary ammonium salts; (**C**) CS reacted with organic acid derivatives to obtain acylated CS; (**D**) carboxymethyl CS was obtained by the reaction of CS with chloroacetic acid under strong alkaline or heating conditions; (**E**) CS reacted with phenol, polyether chain, alkyl chain and other compounds to obtain CS graft copolymer; (**F**) CS reacted with aldehydes or ketones to obtain CSSB. The red part represents the functional groups that undergo chemical reactions.

**Figure 3 foods-13-00439-f003:**
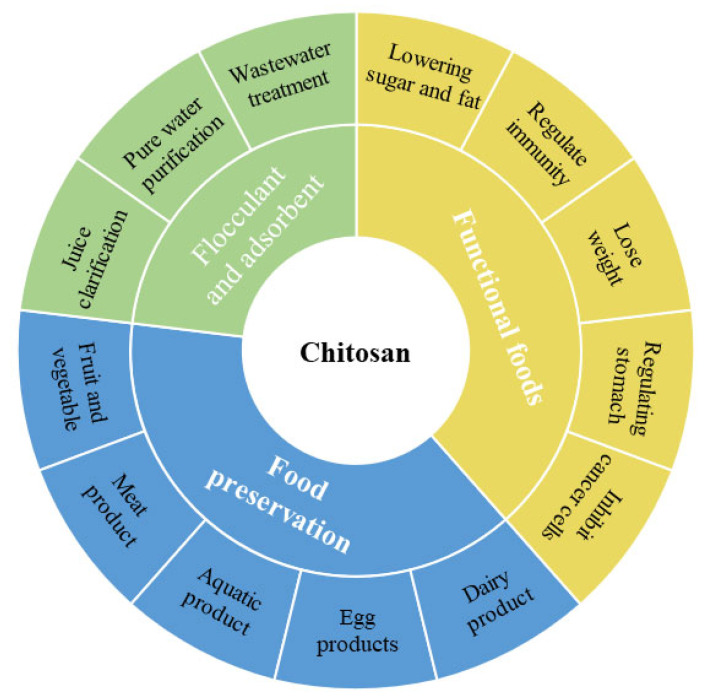
Application of chitosan in the food field.

**Table 1 foods-13-00439-t001:** Comparison of different methods for the extraction of chitosan/chitin.

Source	Deproteinization	Demineralization	Deacetylation	Yield (%)	DD (%)	MW (kDa)	Reference
Acid–base extraction method							
Shrimp	Sodium Hydroxide	Hydrochloric acid	Sodium Hydroxide	15.4	61.24~81.24	1050	[10]
				4.09	73.18~85.26	—	[11]
				54.65	70.9	—	[12]
				—	56.10~88.76	161~451	[13]
				16.4~19.6	96.01~97.2	—	[14]
				22.08	78.4	173	[15]
Crab				30~32.2	83.3~93.3	483~526	[16]
Silkworm pupae				16	66.9	—	[5]
				3.1	85.5	40.9	[17]
Cicada slough				28.2	84.1	37.79	
Mealworm				2.5	85.9	39.75	
Grasshopper				5.7	89.7	39.89	
Shrimp				14.5	91.2	162	
IL extraction method							
Shrimp	[DIPEA][Ac], [DIPEA][P], [DMBA][Ac]	Citric acid		13.40	0.96~1.67	56~160	[18]
Crab	AMIMBr			3.7~12.6	7	70~220	[19]
DES extraction method							
Lobster	Choline chloride, Urea, Glycerol, Malonic aid			4.44~16.19	—	—	[20]
	Choline chloride, Malonic acid			17.21~23.31	5.9~6.7	—	[21]
				19.01~22.21	4.95~5.79	199~312	[22]
Shrimp				13.2	5.54	312	[23]
Strain fermentation process extraction methods							
Shrimp	*Bacillus cereus* SV1	hydrochloric Acid	*Bacillus cereus* SV1	16.55~20	77~59.57	—	[24]
	*Bacillus subtilis*		Sodium Hydroxide	—	74~88	256	[25]
	*Serratia marcescens* B742	*Lactobacillus plantarum* ATCC 8014		18.90	19.83	—	[25]
	*Bacillus amyloliquefaciens*	*Lactobacillus rhamnoides*	—	19.60	—	—	[26]
	*Exiguobacterium proundum*	*Lactobacillus ophilophilus*	—	16.32	3.67	—	[27]
Bio-enzyme extraction methods							
Shrimp shell	Protease-lactic	Lactic acid, acetic acid	—	—	7.33~9.17	—	[28]
	Papain		—	—	19.37	—	[29]
	Alkaline protease		—	15.6	—	—	[30]
	Trypsin	Lactic acid	—	26.85~27.54	17~19	—	[31]

## Data Availability

The original contributions presented in the study are included in the article, further inquiries can be directed to the corresponding author.

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
