# Peer review of "Harnessing the Potential of Chitosan and Its Derivatives for Enhanced Functionalities in Food Applications"

_foods, 2024, doi:10.3390/foods13030439_

Round 1

Reviewer 1 Report

Comments and Suggestions for Authors

The paper entitled “Application of chitosan and its derivatives in the food field” compares various extraction and modification methods used for chitosan recovery from natural sources. The study gradually introduce the classical extraction methods employed at this moment for the production of chitosan. It continuous by presenting the recent advances and recommends improvements that could insure the progress in the field. The study reviews also some of the main application of chitosan in food sector providing insights for future developments. The paper closes with conclusion and highlights possible prospects of chitosan and its derivatives in food field but also in other industries.

The authors will find bellow some corrections and adjustments that should be addressed.

General remarks

-        Even if the abbreviations are easily recognizable (DD, ILs, OD, PC, PVA etc.), their significance should be added at the first appearance in the text.

-        There is an error for the figures and tables numbers.

-        Latin names should be written with italics letters (e.g. lines 479, 585, 600 etc.).

-        Some chemicals are not written correctly (subscripts are sometimes missing – see lines 364-366 for example where also the name of NaBH3CN should be sodium cyanoborohydride; in line 489 the amino group should be written with capital letters etc.).

-        There are phrases rather difficult to understand (e.g. lines 42-44, 146-147, 215-218, 231-234, 238-239, 280-283, 307-308, 317-319, 430 etc.). It is recommended to revise them.

Specific comments:

-        Since the paper is a review, it would be appropriate to specify what was the bibliographic research methodology applied for obtaining the information. What databases were consulted? What was the time period chosen? What were the keywords chosen for the research? What were the criteria on the basis of which some references were included/excluded in/from the review?   

-        In sub-section 2.2.1, lines 269-273, a more elaborated explanation on how the ultrasound would have a positive impact on the fermentation process is required.

-        In sub-section 2.2.2, authors mention the following: in lines 298-300 that “…alkaline proteases have great application potential in the field of chitin extraction. However, the main drawbacks of this method are low chitin yield and poor quality in large-scale production”; in lines 314-316 that “Compared with the traditional chemical extraction method, the immobilization technology of pepsin has a higher protein removal rate”; in lines 332-334 that “Compared with the chemical method, the product of this method has a higher degree of deacetylation, so it has great potential in the related application field of chitosan”. For comparison with the classical extraction method, examples of specific values and references are required.

-        Also in sub-section 2.2.2, in lines 338-340 (“'Glutamic acid-enzymatic hydrolysis ' is a relatively closed process, which has the advantages of mild reaction, greatly reducing the discharge of three wastes and a high comprehensive utilization rate of raw materials”) it is not very clear what are these three wastes.

-        Section 3 should be treated as the previous one, namely including the classical preparation methods of chitosan derivatives and afterwards emphasizing the recently developed processes.

-        It is recommended to divide the sub-section 4.2 in two different ones: one dedicated to the flocculation process, conducted by using chitosan and its derivatives, and another one (placed after the present sub-section 4.3) focused on the treatment of wastewater from food industry.

-        A new section presenting the advantages and drawbacks of using chitosan instead of other natural polymers such alginate for example would complete the review.

Comments on the Quality of English Language

Minor editing of English language required

Reviewer 2 Report

Comments and Suggestions for Authors

1- "Application of chitosan and its derivatives in the food field" is informative and clear, yet it could be strengthened by incorporating specific details or emphasizing the unique advantages of utilizing chitosan and its derivatives in the food industry. For instance: "Harnessing the Potential of Chitosan and Derivatives for Enhanced Functionalities in Food Applications".

2- The abstract succinctly reviews chitosan's application in the food industry, but it would benefit from incorporating specific quantitative data, enhancing the clarity and impact of the presented information. Including key statistics related to extraction efficiency, success rates, or other relevant metrics would provide readers with a more quantifiable understanding of the reviewed studies.

3- The introduction provides a comprehensive overview of the importance of food safety, the shift towards sustainable materials in the food industry, and the potential of chitosan (CS) as a renewable biopolymer. However, it could benefit from a more concise structure to enhance readability. Additionally, incorporating specific examples or quantitative data on the positive environmental impact of CS over traditional materials would strengthen the argument for its adoption. Lastly, a smoother transition between sections, particularly between the discussion of pollution-free extraction methods and the properties of CS, would improve the overall flow of the introduction.

4- What are the current practical limitations of using enzymes, such as alkaline proteases, in large-scale chitin extraction, and how could these challenges be overcome for more widespread adoption?

5- Given the demonstrated effectiveness of CS in preserving various foods, are there ongoing efforts to optimize its application in diverse climates and storage conditions? Furthermore, how can the potential variations in the quality of raw materials, such as fruits and vegetables, impact the performance of CS coatings, and are there strategies to mitigate these variations for consistent preservation outcomes? It would be beneficial to address these critical questions in the article review, shedding light on the collaborative endeavors and technological advancements aimed at refining CS applications across different environmental settings. Understanding how variations in raw material quality might influence CS performance and exploring strategies to ensure uniform preservation outcomes could significantly contribute to the broader discourse on food preservation methods.
